# Spiritual Intelligence as a Mediator between Personality and Emotional and Decisional Forgiveness in Polish Adults

**Justyna Mróz** [1,*], **Kinga Kaleta** [1] **and Katarzyna Skrzypińska** [2]

1    Department of Psychology, Jan Kochanowski University, 25-369 Kielce, Poland
2    Institute of Psychology, University of Gdansk, 80-309 Gdańsk, Poland
*    Correspondence: jmroz@ujk.edu.pl

**Abstract:** Forgiveness is a constructive method of dealing with interpersonal incidents. It is determined by many factors, including personality and spirituality. The purpose of the present study was to explore the relationships between the Big Five personality traits, spiritual intelligence, and emotional and decisional forgiveness in a real situation. The study involved 396 Polish participants aged from 18 to 79. Four tools were used: the Abridged Big Five-Dimensional Circumplex (AB5C), the Emotional Forgiveness Scale (EFS), the Decision to Forgive Scale (DTFS), and the Spiritual Intelligence Self-Report Inventory (SISRI-24). The results demonstrated that personality traits positively correlated with aspects of spiritual intelligence and two types of forgiveness. Personal meaning production (PMP) and transcendental awareness (TA) mediated the relationship between personality and forgiveness, especially when accompanied by taking the decision to forgive. The ability to find meaning and deeper insight into difficult interpersonal incidents was found to be a possible way of linking personality with the forgiveness of others.

**Keywords:** personality; the Big Five; spiritual intelligence; forgiveness; emotional forgiveness; decisional forgiveness

## 1. Introduction

When people are treated unjustly and suffer from a transgression, they may use different methods to cope with the experience (Worthington and Wade 1999; Wade and Worthington 2003). Forgiveness is one of them. Forgiveness is effective and constructive, and it helps offended individuals to overcome their negative states and response to the offence, the offender, and the consequences of the harmful event (Braithwaite et al. 2011; Thompson et al. 2005; Worthington et al. 2019).

Injured individuals can achieve decision-making or emotional forgiveness (Worthington et al. 2007). Decisional forgiveness is understood as a person's intention to control their behaviour toward the offender, not to seek revenge or avoid them, abstain from demonstrating anger, and treat the wrongdoer as a valuable person (Davis et al. 2015). Emotional forgiveness is more than a decision to forgive; as it entails a shift from negative emotions toward the perpetrator to neutral and positive ones, including benevolent thoughts and behaviour (Mróz et al. 2022). People differ in their ability to take the decision to forgive and achieve emotional forgiveness, both at the dispositional level and after a particular harmful episode. One of the most important determinants of their level of forgiveness is personality.

### 1.1. Personality—Forgiveness

Relationships between personality traits and forgiveness have been explored in previous studies (Kaleta and Mróz 2021; Matuszewski and Moroń 2022; Walker 2017). In particular, the Big Five/Five Factors model of personality (Costa and McCrae 1992) has been tested in the context of forgiveness (Apostolou and Demosthenous 2021; Asselmann et al. 2023; Brudek and Kaleta 2023; Dametto and Noronha 2021; Kaleta and Mróz 2019;

Matuszewski and Moroń 2022). The findings have revealed the significance of neuroticism and agreeableness in predicting forgiveness, and less consistently the role of extraversion, conscientiousness, and openness to experience (see meta-analyses by Fehr et al. 2010; Hodge et al. 2020; Riek and Mania 2012).

The studies included various types of forgiveness (forgiveness as a trait and as a state; forgiveness of others, self, and situations), dimensions (positive and negative, decisional and emotional), and samples (general population and clinical samples). Researchers revealed that personality traits and facets were correlated to specific aspects of forgiveness in varied ways. For instance, all facets of neuroticism were negatively related to the negative dimension of forgiveness (reduced negative thoughts, feelings, and behaviour towards the transgressor), whereas merely hostility and vulnerability correlated with the presence of a positive attitude towards the offender (Brose et al. 2005). Additionally, all facets of neuroticism were associated with self-forgiveness and only hostility was related to other-forgiveness (Ross et al. 2004). Agreeableness was positively linked to the trait forgiveness of others in its positive dimension, but not in the negative one (Kaleta and Mróz 2019). Emotional stability positively correlated with emotional forgiveness but not with decisional forgiveness (Kaleta and Mróz 2021).

The above results demonstrate that different dimensions of forgiving involve personality in a complex way. Thus, more in-depth research is required, especially studies explaining the mechanisms linking personality and forgiveness. Few studies included different variables in testing the association between personality and forgiveness and demonstrated the role of gratitude (Neto 2007), spirituality (Leach and Lark 2004), coping (Maltby et al. 2004), or grit (Walker 2017). These studies were mainly correlational and did not explain the underlying mechanisms well enough. Only Koutsos et al. (2008) found that the relationship between agreeableness and the actual act of forgiveness was mediated by the disposition to forgive.

Since research aimed at exploring pathways between personality and forgiveness is limited, the present study is an attempt to fill this gap. When looking for such an indirect path, we are particularly interested in spiritual intelligence (SI).

### 1.2. Personality—Spirituality

Psychologists have long searched for the associations between personality and spirituality. Some authors consider spirituality as a separate human sphere from the physical, mental, and social spheres (e.g., Frankl 1972, 1975). Other researchers point to the psychological basis of spirituality (e.g., Skrzypińska 2022; Socha 2014) and treat it as part of the psyche. There are also those who ponder spirituality to be immanent (Hay 1994), even inborn because it is a human property passed down through genes (e.g., Boyer 2003) or instinct (Bering 2003, 2012). However, correlation studies indicate low links between personality and spirituality (MacDonald 2000; Piedmont 1999; Rose and Exline 2012; Saroglou 2002; Saucier and Skrzypińska 2006; Schnell 2012; Skrzypińska and Chudzik 2017; Skrzypińska and Karasiewicz 2013; Skrzypińska 2022). This is why several researchers in psychology are more in favour of recognizing spirituality as a dimension of personality (Piedmont 1999; MacDonald 2000; Skrzypińska 2014, 2022). Involving personality in spiritual content and activities and the specific features of spirituality and often its original manifestations can be explained from this perspective. One of these manifestations is the human tendency to quest for the meaning of life (Park 2013). Many scholars claim that people's inclination to make sense of the phenomena around them crystallises in this area. At the turn of the millennium, an idea arose that spirituality should have some instrument to search for meaning (Emmons 2000; Zohar and Marshall 2000). This tool is called spiritual intelligence (SI). The initial theoretical research was rather intuitive (Zohar and Marshall 2000), but after 2000, the first attempts to measure this phenomenon began to appear.

Spiritual intelligence (SI) in general is understood as a tool of a mature, shaped personality used to fulfil its spiritual goals or strivings (Emmons 2000; Zohar and Marshall 2000; Halama and Stríženec 2004). Emmons (2000), one of the first SI researchers, defined

SI as competencies and abilities which may be part of a person's expert knowledge and that are significant in problem-solving situations. Emmons (2000) also listed detailed components of SI: (a) the ability to transcend the physical and material world, (b) the capacity to experience heightened states of consciousness, (c) the ability to sanctify everyday experience, (d) the capacity to utilise spiritual resources to solve problems, and (e) the capacity to be virtuous (p. 3).

Many authors assert that SI could be a form of intelligence, involving a set of capacities and abilities that enable people to solve problems and attain goals in their everyday lives. Others strongly deny the possibility of the existence and operation of SI, suggesting the operation of existential intelligence (Gardner 2000). Emmons (2000), answering Gardner's criticism, indicates the *theory of multiple intelligences* and proves that SI fulfils the criteria for an independent intelligence modality (see discussion in Skrzypińska 2020).

King (2008) defines SI as "a set of mental capacities which contribute to the awareness, integration, and adaptive application of the nonmaterial and transcendent aspects of one's existence, leading to such outcomes as deep existential reflection, enhancement of meaning, recognition of a transcendent self, and mastery of spiritual states" (p. 56). The author derives variables that are easy to operationalise. Importantly, King adopted an empirical approach that revealed four SI factors: Critical Existential Thinking (CET), Personal Meaning Production (PMP), Transcendental Awareness (TA), and Conscious State Expansion (CSE) (King 2008, pp. 56, 61). CET concerns the capacity to critically contemplate meaning, purpose, and other existential/metaphysical issues (e.g., existence, reality, death, the universe); to come to original existential conclusions or philosophies; and to contemplate non-existential issues concerning one's existence (i.e., from an existential perspective). PMP means the ability to derive personal meaning and purpose from all physical and mental experiences, including the capacity to create and master/live according to a life purpose. TA is the capacity to identify transcendent dimensions/patterns of the self (i.e., a transpersonal or transcendent self), of others, and the physical world (e.g., holism, non-materialism) during normal states of consciousness, accompanied by the capacity to identify their relationship to one's self and the physical world. Finally, CSE refers to the ability to enter and exit higher/spiritual states of consciousness (e.g., pure consciousness, cosmic consciousness, unity, oneness) at one's discretion (as in deep contemplation or reflection, meditation, prayer, etc.).

Several scholars undertook to check whether SI understood this way operates in different cultures (e.g., Antunes et al. 2018; Atroszko et al. 2021; Chan and Siu 2016). Some of them also decided to check whether personality/spirituality can use SI to achieve a certain goal (Skrzypińska et al. 2019). We were particularly interested in forgiveness as a variable that facilitates social functioning.

### 1.3. Spirituality—Forgiveness

Although spiritual intelligence and spirituality are different constructs, they share some similarities, as they both refer to humans' internal lives of the spirit and mind as well as making sense of the phenomena in the external world. As a result, it is reasonable to support our study with research on spirituality in general and in its dimensions (Mróz et al. 2021). Various aspects of spirituality have been associated with forgiveness. Spirituality demonstrated a positive relationship with the dispositional forgiveness of others in students (Koutsos et al. 2008). Existential well-being was positively related to the trait forgiveness in adults aged from 50 to 95 years (Lawler-Row and Piferi 2006). Spiritual experiences were positively correlated with the traits forgiveness of self and forgiveness of others in participants of addiction treatment programs (Lyons et al. 2011). Additionally, spiritual experience, religious identity, and positive spiritual/religious coping were positively linked to the tendency to forgive in patients with advanced chronic heart failure (Park et al. 2014). In turn, spiritual disappointment negatively correlated with forgivingness (Sandage and Williamson 2010). Taking state forgiveness into consideration, spiritual commitment and positive attitudes towards the sacred or other sources of spirituality negatively

correlated with unforgiveness, whereas anger towards God, sacred loss, and desecration were positively linked to unforgiveness in students (Choe et al. 2016). In addition, spiritual transcendence positively correlated with benevolent motivation and inversely with revenge and avoidance (Edara 2015). Spiritual intelligence, however, differs from above dimensions of spirituality. SI is a meaning-searching tool (Emmons 2000; Zohar and Marshall 2000) that consists of cognitive abilities and competencies (King 2008), not spiritual well-being, experience, commitment or beliefs. Thus, it should be measured directly. When it comes to previous studies, all aspects of SI revealed positive associations with the multidimensional trait forgiveness (Mróz et al. 2022). Additionally, two facets of SI, personal meaning production (PMP) and transcendental awareness (TA) demonstrated a negative association with revenge and a positive correlation with benevolent motivation. In addition, the trait forgiveness was a mediator between spiritual intelligence dimensions (personal meaning production and transcendental awareness) and episodic forgiveness, measured as revenge and benevolent motivation.

Summing up, significant relationships between personality, spirituality, and forgiveness have been found, but most of them were not tested in one piece of research. Only (Leach and Lark 2004) aimed at examining whether spirituality (measured as spiritual well-being and spiritual transcendence) explained variance in dispositional forgiveness beyond that of personality. They showed that spirituality explained a significant portion of the variance, but mainly in only one factor. Thus, our first goal is to explore the relationships between personality, spirituality, and forgiveness at a decisional and emotional level. Moreover, as no study to date tested a more complex model including the three variables, we are interested in how personality and spirituality may enhance the actual forgiveness of others. Hence, the second goal of the study is to investigate whether SI plays a mediating role in the relationship between the five factors of personality and emotional and decisional forgiveness (see Figure 1).

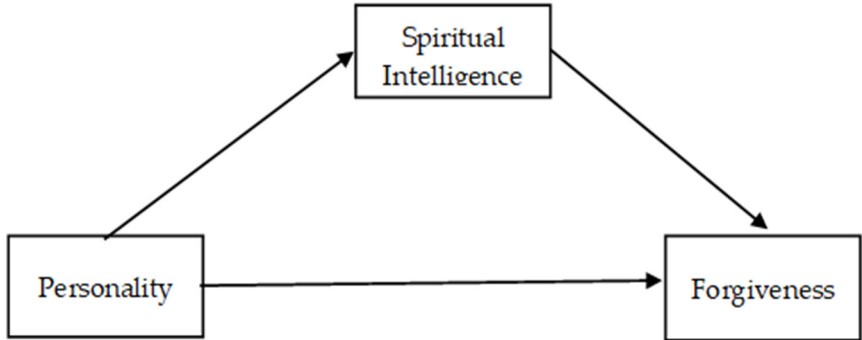

**Figure 1.** A model of the relationship between personality and forgiveness with spiritual intelligence as a mediator.

## 2. Materials and Methods

### 2.1. Power Analysis

To determine the minimum sample size for the current study, G*Power version 3.1. Program (Faul et al. 2007) was used. The sample size required for multiple regression analyses with eleven independent variables for detecting a medium effect ($f2 = 0.03$) with a power of 0.80 and a 0.05 level of significance was $N = 208$ or more.

### 2.2. Participants and Procedure

The study participants were adults aged 18 to 79 years ($M = 28.93$; $SD = 12.32$) from Poland. The sample comprised 396 respondents, of whom 273 (68.9%) were female, 122 (20.8%) were male, and one person indicated "other" as their gender. Of these, 39.4% (n = 156) completed university education, 55.6% completed secondary education (n = 220), and 5.1% (n = 20) completed primary education. They usually lived in cities (45.5%; n = 178) or in the country (37.9%; n = 150), and less often in towns (16.7%; n = 66). The respondents

were requested to participate in the study voluntarily—no remuneration was offered to them. The study was conducted in accordance with the principles of the Declaration of Helsinki relating to ethical principles for research with human participants.

*2.3. Methods*

Big Five-Dimensional Circumplex (AB5C; Hofstee et al. 1992; Skrzypińska and Ladd 2018) was used to assess five personality domains. The AB5C scale consists of 30 items (adjectives) (e.g., Extraversion—talkative; Agreeableness—cheerful). Participants indicate their answers on 7-point Likert scales, where higher numbers reflect more agreement with the content of each item (1—*strongly disagree* to 7—*strongly agree*). In the current sample, the Cronbach's alpha were between 0.55 (Intellect) and 0.80 (Agreeableness).

Emotional Forgiveness Scale—EFS (Hook et al. 2012; Mróz et al. 2022) was used to assess the intensity of emotional forgiveness and peace of mind in relation to a particular offence. The measure consists of 8 items (e.g., *I feel sympathy toward him or her*) rated on a 5-point Likert scale (1—*strongly disagree* to 5—*strongly agree*). Four items measure positive emotions toward the offender (EFS-PP), and four items measure reduced negative emotions toward the offender (EFS-RN). For the current sample, Cronbach's alpha coefficient for the EFS was 0.88.

Decision to Forgive Scale (DTFS; Davis et al. 2015; Mróz et al. 2022) was used to assess decisional forgiveness. The DTFS consists of 5 items (e.g., *I made a commitment to forgive him or her*) rated on a 5-point Likert scale (1—*strongly disagree* to 5—*strongly agree*). In the present study, Cronbach's alpha for DTFS was 0.91.

Spiritual Intelligence Self-Report Inventory (SISRI-24) (King 2008; Atroszko et al. 2021) was used to assess spiritual intelligence. The SISRI consists of 24 items (e.g., *My ability to find meaning and purpose in life helps me adapt to stressful situations*). The scale has four subscales: CET (Critical Existential Thinking), PMP (Personal Meaning Production), TA (Transcendental Awareness), and CSE (Conscious State Expansion) (King and DeCicco 2009). Participants rate each item on a 5-point Likert scale (1—*not at all true of me* to 5—*completely true of me*). To interpret the results, this finding should be taken into account and analysed in relation to individual scales, and not the general SI factor. In the current sample, Cronbach's alpha values were as follows: CET 0.78, PMP 0.77, TA 0.81, and CSE 0.88.

*2.4. Data Analysis*

The data were evaluated for potential errors in the expected range of values and for any indicators of careless answers. We used the Mahalanobis distance to evaluate the outliers (De Maesschalck et al. 2000). All results fulfilled the criteria. We also examined the common method bias adopting Harman's single-factor test, which was launched by factor analysis without rotation (Podsakoff et al. 2003). The principal component analysis revealed that the first factor accounted for 28.25% of the variance. For all the data, we used the Kolmogorov–Smirnov test to verify the hypothesis of a normal distribution. The $p$ values were below 0.05 for all tools. Its results indicated that the data are not normally distributed. Spearman's rho correlations were used to estimate the relationship between the main variables: personality, spiritual intelligence, and forgiveness. We used IBM SPSS software (version 26, PS IMAGO PRO 6.0, Predictive Solutions) and PROCESS (Hayes 2017) to test the proposed mediating model (model 4). The PROCESS was estimated with 5000 bootstrap samples and 95% bias-corrected bootstrap intervals for all indirect effects.

**3. Results**

Table 1 presents correlations (Spearman's rho) between the variables. Data show extraversion was positively related to personal meaning production (PMP) and conscious state expansion (CSE). Agreeableness was positively associated with PMP, TA (transcendental awareness), CSE, and both decisional and emotional forgiveness. Conscientiousness was positively related to PMP, TA, and CSE. Emotional stability was positively related

to PMP, CSE, and emotional forgiveness. Intellect is positively related to CET (critical existential thinking), PMP, TA, CSE, and the reduction in negative emotions. Additionally, PMP was positively correlated with emotional and decisional forgiveness. TA was positively linked to decisional forgiveness. CSE was positively associated with the reduction in negative emotions.

To examine whether the four subscales of spiritual intelligence (critical existential thinking, personal meaning production, transcendental awareness, conscious state expansion) mediated the association between personality and decisional and emotional forgiveness, we used a multiple mediation model (Model 4 in PROCESS). As forgiveness might be age- and gender-specific (Kaleta and Mróz 2018, 2022; Cabras et al. 2022), age and gender were used as covariates. Indirect effects are presented in Tables 2 and 3.

Agreeableness was positively associated with personal meaning production ($\beta = 0.030$, $p < 0.001$), transcendental awareness ($\beta = 0.09$, $p < 0.043$), and conscious state expansion ($\beta = 0.10$, $p < 0.046$), while only personal meaning production ($\beta = 0.14$, $p < 0.027$) was associated with emotional forgiveness ($R^2 = 0.039$, $F(382,7) = 2.150$, $p < 0.037$). On the other hand, personal meaning production ($\beta = 0.14$, $p < 0.025$) and transcendental awareness ($\beta = 0.26$, $p < 0.006$) were significantly related to decisional forgiveness ($R^2 = 0.06$, $F(382,7) = 3.527$, $p < 0.001$).

The indirect effect of agreeableness on emotional forgiveness via spiritual intelligence was found to be significant ($\beta = 0.053$, 95% CI [0.011, 0.100]). However, only PMP was a significant mediator ($\beta = 0.048$, 95% CI [0.004, 0.098]). The indirect effect accounted for 39% of the total effect.

The indirect effect of agreeableness on decisional forgiveness through spiritual intelligence was found to be significant ($\beta = 0.054$, 95% CI [0.020, 0.095]). PMP ($\beta = 0.04$, 95% CI [0.002, 0.083]) and TA ($\beta = 0.02$, 95% CI [0.000, 0.047]) were significant mediators. The indirect effect accounted for 48.49% of the total effect.

Extraversion was positively associated with PMP ($\beta = 0.16$, $p < 0.001$) and CS ($\beta = 0.11$, $p < 0.02$), while PMP ($\beta = 0.18$, $p < 0.01$) was associated with emotional forgiveness ($R^2 = 0.04$, $F(382,7) = 2.07$, $p < 0.05$), and PMP ($\beta = 0.17$, $p < 0.01$) and TA ($\beta = 0.20$, $p < 0.01$) with decisional forgiveness ($R^2 = 0.06$, $F(382,7) = 3.315$, $p < 0.001$).

The indirect effect of extraversion on emotional forgiveness via spiritual intelligence was found to be significant ($\beta = 0.031$, 95% CI [0.005, 0.056]). PMP ($\beta = 0.029$, 95% CI [0.005, 0.055]) was a significant mediator. The indirect effect represented 50.1% of the total effect.

The indirect effect of extraversion on decisional forgiveness via spiritual intelligence was significant ($\beta = 0.03$, 95% CI [0.001, 0.061]). PMP ($\beta = 0.03$, 95% CI [0.005, 0.059] was a significant mediator. The indirect effect represented 56.8% of the total effect.

Conscientiousness was positively associated with PMP ($\beta = 0.24$, $p < 0.001$), and CS ($\beta = 0.17$, $p < 0.001$), while PMP ($\beta = 0.17$, $p < 0.01$), and TA ($\beta = 0.20$, $p < 0.01$) was associated with decisional forgiveness ($R^2 = 0.07$, $F(382,9) = 3.28$, $p < 0.01$).

The indirect effect of conscientiousness on decisional forgiveness via spiritual intelligence was significant ($\beta = 0.05$, 95% CI [0.016, 0.089]). PMP ($\beta = 0.04$, 95% CI [0.008, 0.079]) was a significant mediator. The indirect effect represented 30% of the total effect.

The mediation model of conscientiousness on emotional forgiveness via spiritual intelligence was not significant.

Intellect was positively related to CET ($\beta = 0.20$, $p < 0.001$), PMP ($\beta = 0.24$, $p < 0.001$), TA ($\beta = 0.27$, $p < 0.001$), and CS ($\beta = 0.27$, $p < 0.001$). PMP ($\beta = 0.17$, $p < 0.01$) and TA ($\beta = 0.21$, $p < 0.01$) were linked to decisional forgiveness ($R^2 = 0.06$, $F(382,7) = 3.32$, $p < 0.01$).

The indirect effect of intellect on decisional forgiveness via spiritual intelligence was significant ($\beta = 0.04$, 95% CI [0.007, 0.096]). PMP ($\beta = 0.04$, 95% CI [0.008, 0.081]) and TA ($\beta = 0.05$, 95% CI [0.015, 0.111]) were significant mediators. The indirect effect accounted for 38% of the total effect.

The indirect effect of intellect on emotional forgiveness via spiritual intelligence was not significant. The indirect effects of emotional stability on emotional and decisional forgiveness were not significant.

**Table 1.** Descriptive Statistics, and Bivariate Correlations Between Variables.

| | | 1 | 2 | 3 | 4 | 5 | 6 | 7 | 8 | 9 | 10 | 11 | 12 | 13 |
|---|---|---|---|---|---|---|---|---|---|---|---|---|---|---|
| 1 | Extraversion | - | | | | | | | | | | | | |
| 2 | Agreeableness | 0.37 ** | - | | | | | | | | | | | |
| 3 | Conscientiousness | 0.35 ** | 0.41 ** | - | | | | | | | | | | |
| 4 | Emotional Stability | 0.36 ** | 0.42 ** | 0.54 ** | - | | | | | | | | | |
| 5 | Intellect | 0.38 ** | 0.36 ** | 0.31 ** | 0.36 ** | - | | | | | | | | |
| 6 | SISRI_CET | −0.01 | 0.01 | −0.03 | −0.06 | 0.24 ** | - | | | | | | | |
| 7 | SISRI_PMP | 0.19 ** | 0.36 ** | 0.28 ** | 0.31 ** | 0.28 ** | 0.34 ** | - | | | | | | |
| 8 | SISRI_TA | 0.06 | 0.17 ** | 0.11 * | 0.00 | 0.31 ** | 0.67 ** | 0.53 ** | - | | | | | |
| 9 | SISRI_CSE | 0.13 ** | 0.14 ** | 0.18 ** | 0.21 ** | 0.31 ** | 0.52 ** | 0.53 ** | 0.56 ** | - | | | | |
| 10 | EFS | −0.02 | 0.12 * | 0.07 | 0.12 * | 0.01 | 0.00 | 0.15 ** | 0.07 | 0.05 | - | | | |
| 11 | EFS_PP | −0.05 | 0.05 | 0.06 | 0.00 | −0.06 | 0.01 | 0.06 | 0.06 | 0.00 | 0.80 ** | - | | |
| 12 | EFS_RN | 0.06 | 0.15 ** | 0.06 | 0.19 ** | 0.10 * | 0.01 | 0.21 ** | 0.08 | 0.11 * | 0.72 ** | 0.16 ** | - | |
| 13 | DTFS | 0.00 | 0.12 * | 0.03 | 0.01 | 0.01 | 0.01 | 0.18 ** | 0.15 ** | 0.05 | 0.66 ** | 0.59 ** | 0.40 ** | - |
| | *M* | 26.54 | 32.56 | 28.04 | 24.87 | 27.02 | 19.18 | 16.90 | 21.20 | 12.70 | 22.52 | 11.00 | 11.52 | 16.87 |
| | *SD* | 6.66 | 5.27 | 5.68 | 5.86 | 4.93 | 6.20 | 3.89 | 4.72 | 4.75 | 6.04 | 4.23 | 3.67 | 4.97 |

Note. * $p < 0.05$; ** $p < 0.01$;; **Abbreviations:** SISRI—Spiritual Intelligence Self-Report Inventory with its scales: CET—Critical Existential Thinking, PMP—Personal Meaning Production, TA—Transcendental Awareness, CSE—Conscious State Expansion. EFS—Emotional Forgiveness Scale (total score), EFS_PP—Presence of Positive Emotions subscale, EFS_RN—Reduction in Negative Emotions subscale, DTFS—Decision to Forgive Scale.

**Table 2.** Standardised indirect effects—standard errors (SE)—and 95% confidence intervals (CI) for emotional forgiveness.

| Model Pathways | Effect | SE | 95% CI | |
| --- | --- | --- | --- | --- |
| | | | Lower | Upper |
| Agreeableness → SI → EF (total IE) | 0.04 | 0.02 | 0.012 | 0.092 |
| Agreeableness → PMP → EF | 0.04 | 0.02 | 0.004 | 0.087 |
| Agreeableness → CET → EF | 0.00 | 0.00 | −0.008 | −.014 |
| Agreeableness → TA → EF | 0.00 | 0.01 | −0.009 | 0.029 |
| Agreeableness → CES → EF | 0.00 | 0.01 | −0.021 | 0.015 |
| Extraversion → SI → EF (total IE) | 0.03 | 0.01 | 0.004 | 0.062 |
| Extraversion → PMP → EF | 0.03 | 0.01 | 0.005 | 0.060 |
| Extraversion → CET → EF | 0.00 | 0.00 | −0.008 | 0.017 |
| Extraversion → TA → EF | 0.00 | 0.00 | −0.008 | 0.015 |
| Extraversion → CES → EF | 0.00 | 0.01 | −0.021 | 0.017 |
| Conscientiousness → SI → EF (total IE) | 0.04 | 0.01 | 0.007 | 0.081 |
| Conscientiousness → PMP → EF | 0.04 | 0.02 | 0.007 | 0.077 |
| Conscientiousness → CET → EF | 0.00 | 0.00 | −0.007 | 0.025 |
| Conscientiousness → TA → EF | 0.00 | 0.00 | −0.007 | 0.025 |
| Conscientiousness → CES → EF | 0.00 | 0.01 | −0.032 | 0.020 |
| Emotional Stability → SI → EF | 0.03 | 0.02 | −0.008 | 0.088 |
| Emotional Stability → PMP → EF | 0.04 | 0.02 | 0.005 | 0.092 |
| Emotional Stability → CET → EF | 0.00 | 0.00 | −0.011 | 0.015 |
| Emotional Stability → TA → EF | 0.00 | 0.00 | −0.010 | 0.016 |
| Emotional Stability → CES → EF | −0.01 | 0.01 | −0.046 | 0.024 |
| Intellect → SI → EF (total IE) | 0.03 | 0.02 | −0.002 | 0.077 |
| Intellect → PMP → EF | 0.04 | 0.01 | 0.010 | 0.079 |
| Intellect → CET → EF | −0.01 | 0.01 | −0.062 | 0.011 |
| Intellect → TA → EF | 0.01 | 0.02 | −0.022 | 0.068 |
| Intellect → CES → EF | −0.00 | 0.02 | −0.047 | 0.034 |

Abbreviations: SI—Spiritual Intelligence Self-Report Inventory with its scales: CET—Critical Existential Thinking, PMP—Personal Meaning Production, TA—Transcendental Awareness, CSE—Conscious State Expansion, EF—emotional forgiveness.

**Table 3.** Standardised indirect effects—standard errors (SE)—and 95% confidence intervals (CI) for decisional forgiveness.

| Model Pathways | Effect | SE | 95% CI | |
| --- | --- | --- | --- | --- |
| | | | Lower | Upper |
| Agreeableness → SI → DF (total IE) | 0.05 | 0.02 | 0.021 | 0.102 |
| Agreeableness → PMP → DF | 0.04 | 0.02 | 0.002 | 0.090 |
| Agreeableness → CET → DF | 0.00 | 0.00 | −0.008 | 0.018 |
| Agreeableness → TA → DF | 0.02 | 0.01 | 0.000 | 0.050 |
| Agreeableness → CES → DF | −0.01 | 0.01 | −0.028 | 0.005 |
| Extraversion → SI → DF (total IE) | 0.02 | 0.01 | −0.001 | 0.060 |
| Extraversion → PMP → DF | 0.02 | 0.01 | 0.004 | 0.058 |
| Extraversion → CET → DF | 0.00 | 0.00 | −0.008 | 0.020 |
| Extraversion → TA → DF | 0.00 | 0.01 | −0.015 | 0.031 |
| Extraversion → CES → DF | −0.01 | 0.01 | −0.030 | 0.006 |
| Conscientiousness → SI → DF (total IE) | 0.05 | 0.01 | 0.014 | 0.090 |
| Conscientiousness → PMP → DF | 0.04 | 0.01 | 0.007 | 0.078 |
| Conscientiousness → CET → DF | 0.00 | 0.00 | −0.008 | 0.023 |
| Conscientiousness → TA → DF | 0.01 | 0.01 | −0.002 | 0.047 |
| Conscientiousness → CES → DF | −0.01 | 0.01 | −0.040 | 0.009 |
| Emotional Stability → SI → DF | 0.04 | 0.02 | −0.003 | 0.096 |
| Emotional Stability → PMP → DF | 0.05 | 0.02 | 0.009 | 0.102 |
| Emotional Stability → CET → DF | 0.00 | 0.00 | −0.011 | 0.022 |
| Emotional Stability → TA → DF | 0.00 | 0.01 | −0.016 | 0.029 |
| Emotional Stability → CES → DF | −0.01 | 0.01 | −0.054 | 0.013 |
| Intellect → SI → DF (total IE) | 0.05 | 0.02 | 0.006 | 0.097 |
| Intellect → PMP → DF | 0.04 | 0.01 | 0.009 | 0.080 |
| Intellect → CET → DF | −0.02 | 0.01 | −0.067 | −0.001 |
| Intellect → TA → DF | 0.05 | 0.02 | 0.014 | 0.110 |
| Intellect → CES → DF | −0.02 | 0.01 | −0.060 | 0.018 |

Abbreviations: SI—Spiritual Intelligence Self-Report Inventory with its scales: CET—Critical Existential Thinking, PMP—Personal Meaning Production, TA—Transcendental Awareness, CSE—Conscious State Expansion, DF—decisional forgiveness.

## 4. Discussion

The present study aimed at exploring the relationships between personality, spiritual intelligence and decisional and emotional forgiveness in the context of a particular offence. We found positive associations between the variables. All personality traits correlated with dimensions of SI. Agreeableness, emotional stability, and intellect were related to a reduction in negative emotions (dimension of emotional forgiveness), and agreeableness was linked to decisional forgiveness. All correlations were positive: individuals with higher scores in specific personality traits reported higher SI and forgiveness. Additionally, selected dimensions of SI were positively related to emotional and decisional forgiveness.

The findings are consistent with the previous research demonstrating that personality is related to spiritual intelligence (e.g., Amrai et al. 2011; Mahasneh et al. 2015; Skrzypińska et al. 2019) and emotional and decisional forgiveness (Kaleta and Mróz 2021). We additionally found that personality is associated with the reduction in negative emotions following a transgression and the decision to forgive, but not with the presence of positive emotions. Thus, some personality traits help offended individuals to reduce the negative experience after being hurt and control their behaviour, but they do not determine any positive regard towards the offender. The latter might require more effort and involve additional psychological processes (Mróz and Kaleta 2017). Approach motivation towards the offender is more complicated than just reducing negative motives which usually comes with time (McCullough et al. 2003). Another explanation could be "grief burnout": perhaps this negative emotion consumes so many resources that it is difficult to generate positive emotions. It seems that the perpetrator and the amount of harm suffered may play an important role here as well. It would probably be worth controlling resilience also as an important personality variable related to the process of adaptation to difficult situations in the future. The present study also revealed associations between SI and emotional and decisional forgiveness that have not been tested before. We found that the higher the ability to derive personal meaning (PMP), experience extended states of consciousness (CSE), and identify transcendent dimensions of life (TA), the better the reduction in negative emotions following a transgression and the stronger the decision to forgive. Previous cross-cultural studies also showed correlations of these three SI elements with the sense of meaning in life: high with PMP and average with TA and CSE (Skrzypińska et al. 2019). Our findings demonstrated how these specific mental capacities favour particular dimensions of actual forgiveness. In the future, the sense of the meaning of life, which may become real during an act of forgiveness, could be included in the analyses.

Our study also revealed the mediating role of SI in the relationship between personality and forgiveness. The ability to derive personal meaning from different experiences (PMP) might be a way of linking agreeableness, extraversion, conscientiousness, and intellect with forgiveness, especially in its decisional dimension. Additionally, agreeableness and intellect were positively related to the capacity to identify transcendent dimensions of the self, others, and the physical world (TA) which in turn was associated with a stronger decision to forgive. These findings indicate correlations between personality and SI factors that suggest it is possible to use this instrument to achieve goals such as forgiveness. Personality encompasses various aspects and abilities, including spirituality, and different ways of thinking and interpreting particular experiences, including tough ones. Our findings show that the more agreeable, extroverted, conscientious, or open-minded people are, the more they find meaning and deeper insight in difficult interpersonal incidents, which in turn helps them to achieve forgiveness: a way to cope with the transgression (Worthington and Scherer 2004). At the same time, this indicates the protective function of the collaboration between personality and SI. Our results are consistent with previous research reporting that SI is positively associated with constructive coping strategies (Safavi et al. 2018) and inversely related to stress, anxiety, and depression (Du et al. 2013; Safavi et al. 2018; Salmabadi et al. 2016) which results from unforgiveness. At the same time, it should be emphasised that SI does not enter into relationships with rational intelligence (RI), but it cooperates with emotional intelligence (EI) (Skrzypińska et al. 2019). In the future, when designing new research,

but also in diagnostics and therapy, this should be taken into account: it is not rational intelligence, but spiritual and emotional intelligence that can facilitate adaptation in life (Sánchez-Álvarez et al. 2016; Trigueros et al. 2020; Zeidner et al. 2006).

The present study extends cross-cultural support for the SI model by providing further findings from the Polish sample, alongside Chinese (Chan and Siu 2016), Portuguese (Antunes et al. 2018), Greek (Polemikou et al. 2019), Latvian (Grasmane et al. 2022) or New Zealand (O'Sullivan and Lindsay 2022). Moreover, the studies in question supported the notion that personality/spirituality can use SI to achieve certain goals (Skrzypińska et al. 2019). Findings demonstrated that SI is helpful in gaining resilience or coping with the transgression through forgiveness, but not necessarily in enhancing wellbeing (Mróz et al. 2021; O'Sullivan and Lindsay 2022).

## 5. Limitations

Our research is not free from limitations. The first one is the cross-sectional design of the study using self-report questionnaires that prevent us from reaching conclusions about the direction of the observed effects. Another limitation is the lack of control for situational factors related to a particular offence. Acts of forgiveness are predicted mainly by an apology, the severity of the offence, and the relationship satisfaction (Kaleta and Mróz 2021). Further studies should involve such contextual factors, because the models we tested are not comprehensive enough. Third, our sample was mostly female. Despite the fact that previous research has not demonstrated gender differences in spiritual intelligence (Pant and Srivastava 2019), or has been inconsistent with forgiveness (Cabras et al. 2022; Kaleta and Mróz 2022), it is worthwhile for future research to address the gender imbalance, and to examine whether there are differences in the relationship between spiritual intelligence and forgiveness in females and males. Finally, future research examining associations between personality, spiritual intelligence, and episodic forgiveness might apply alternative models, including other mediators, for instance, emotional intelligence (Keaten and Kelly 2008; Sliter et al. 2013). Perhaps it would also be possible to check the mediation analysis, where the sense of life would be the explained variable, and forgiveness would be the mediator, preceded by the output variable, i.e., personality. Performing a path analysis in this case could reveal a much more complicated mechanism of forgiveness than the one presented in this article.

**Author Contributions:** Conceptualization, J.M. and K.K.; methodology, J.M. and K.K.; validation, J.M. and K.K.; formal analysis, J.M.; investigation, J.M., K.K. and K.S.; resources, J.M. and K.K.; data curation, J.M.; writing—original draft preparation, J.M., K.K., K.S.; writing—review and editing, J.M., K.K., K.S.; visualization, J.M.; supervision, J.M.; project administration, J.M. All authors have read and agreed to the published version of the manuscript.

**Funding:** This research received no external funding.

**Institutional Review Board Statement:** The study was conducted in accordance with the Declaration of Helsinki. Participation in the study was anonymous and voluntary.

**Informed Consent Statement:** Informed consent was obtained from all subjects involved in the study.

**Data Availability Statement:** The data that support the findings of this study are available from the corresponding author [J.M.] upon request.

**Conflicts of Interest:** The authors declare no conflict of interest.

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
