# Peer review of "Spiritual Intelligence as a Mediator between Personality and Emotional and Decisional Forgiveness in Polish Adults"

_religions, doi:10.3390/rel14050574_

Round 1
Reviewer 1 Report
1. The title does not indicate the country/nation/group that this research concerns. It is worth adding this information to the title of the article.
2. The summary is generally written in a correct way, but there is no information about the research participants’ country/nationality.
3. Introduction is very well-written. The contents are consistent, logical and ordered in a clear way. The presented review of previous research takes into consideration the multifaceted qualities of the analysed phenomena as well as of the viewpoints presented in the subject literature.
4. In the description of the research group (2.2. Participants and procedure) there was no mention of the country and time when the research was carried out. It is necessary to add this information to the article.
5. In the description of AB5C scale the values of the extreme responses in Likert scale (from 1-…to 7-…) and Cronbach’s alpha were not specified. The description of the remaining methods is comprehensive.
6. Results are correctly described and presented.
7. Discussion was well-conducted.
8. The article is cognitively valuable and I recommend it for publication after minor revision and after correcting it accordingly to the remarks included in this review.
Author Response
Reviewer #1
First of all, we would like to thank the Reviewers for the time devoted to reading our article. We would also like to thank you for your valuable comments and useful tips, which certainly improve the quality of our study. The responses of the authors to the suggestions of the Reviewers are marked in red.
- The title does not indicate the country/nation/group that this research concerns. It is worth adding this information to the title of the article.
We have added information about the nationality of the participants’ nation in the title.
- The summary is generally written in a correct way, but there is no information about the research participants’ country/nationality.
We have completed this information.
- Introduction is very well-written. The contents are consistent, logical and ordered in a clear way. The presented review of previous research takes into consideration the multifaceted qualities of the analysed phenomena as well as of the viewpoints presented in the subject literature.
- In the description of the research group (2.2. Participants and procedure) there was no mention of the country and time when the research was carried out. It is necessary to add this information to the article.
We have included information about on the country of origin of the research participants
- In the description of AB5C scale the values of the extreme responses in Likert scale (from 1-…to 7-…) and Cronbach’s alpha were not specified. The description of the remaining methods is comprehensive.
We have added this information.
- Results are correctly described and presented.
- Discussion was well-conducted.
- The article is cognitively valuable and I recommend it for publication after minor revision and after correcting it accordingly to the remarks included in this review.
Reviewer 2 Report
The study is meaningful as it enhances understanding on the effects of Spiritual Intelligence on Personality and Forgiveness. The study’s objectives, theoretical framework, methods, results, and limitations are logically presented. Practical and meaningful suggestions on future research are also provided. The authors may consider the suggestions listed in the review report.

Author Response
Reviewer #2
First of all, we would like to thank the Reviewers for the time devoted to reading our article. We would also like to thank you for your valuable comments and useful tips, which certainly improve the quality of our study. The responses of the authors to the suggestions of the Reviewers are marked in red.
- The use of literatures is highly relevant for supporting the argument, but more recent studies, especially on relationship between personality and forgiveness, may be included.
We have included several newest references referring to the personality-forgiveness association, such as:
Apostolou, Menelaos, and Andriana Demosthenous. 2021. ‘Why People Forgive Their Intimate Partners’ Infidelity: A Taxonomy of Reasons’. Adaptive Human Behavior and Physiology 7: 54-71. https://doi.org/10.1007/s40750-020-00153-1.
Asselmann, Eva, Elke Holst, and Jule Specht. 2023. ‘Longitudinal Bidirectional Associations between Personality and Becoming a Leader’. Journal of Personality 91 (2): 285-298. https://doi.org/10.1111/jopy.12719.
Brudek, Paweł, and Kinga Kaleta. 2023. ‘Marital Offence-specific Forgiveness as Mediator in the Relationships between Personality Traits and Marital Satisfaction Among Older Couples: Perspectives on Lars Tornstam's Theory of Gerotranscendence’. Ageing & Society 43 (1): 161-179. https://doi.org/10.1017/S0144686X21000465.
Dametto, Denise Martins, and Ana Paula Porto Noronha. 2021. ‘Study between Personality Traits and Character Strengths in Adolescents’. Current Psychology 40: 2067-2072. https://doi.org/10.1007/s12144-019-0146-2.
Matuszewski, Krzysztof, and Marcin Moroń. 2022. ‘The HEXACO Model of Personality, Religiosity, and Trait Forgiveness’. Pastoral Psychology 71 (4): 525-543. https://doi.org/10.1007/s11089-022-01006-2.
These are the most recent studies testing the relationships between personality traits and forgiveness. They measured the analysed variables, however, at the overall level, not in their facets or dimensions. As a result, we could include them as examples of research in question, but we were not able to complete our more comprehensive analyses (in one of the paragraphs in the introduction) which rely on the studies published slightly earlier, but still current and valid.
- Studies on spirituality and personality, and spirituality and forgiveness are used to support the relationship between spiritual intelligence and personality, and spiritual intelligence and forgiveness. Although spiritual intelligence and spirituality share similarities, they are two different concepts, it would be better if more explanation could be offered on how these two concepts are different and similar and why studies on spirituality could be applied to spiritual intelligence.
We completely agree with the Reviewer’s opinion that spirituality and spiritual intelligence are two different concepts, including similar and different qualities. We have included a brief clarification in paragraph 1.3. Spirituality - forgiveness:
Although spiritual intelligence and spirituality are different constructs, they share some similarities, as they both refer to humans’ internal lives of the spirit and mind as well as making sense of the phenomena in the external world. As a result, it is reasonable to support our study with research on spirituality in general and in its dimensions (Mróz, Kaleta, and Skrzypińska 2021). [(...])
Spiritual intelligence, however, differs from above dimensions of spirituality. SI is a meaning-searching tool (Emmons 2000; Zohar, Marshall, and Marshall 2000) that consists of cognitive abilities and competencies (King 2008), not spiritual well-being, experience, commitment or beliefs. Thus, it should be measured directly.
- On p. 3, “Several scholars undertook to check whether SI……SI to achieve a certain goal (Skrzypinska et al. 2019)”. It would be better if clearer discussion on how the studies mentioned in this paragraph relevant to the aims of the present studies.
As recommended above, we added an extra paragraph:
The present study extends cross-cultural support for the SI model by providing further findings from the Polish sample, alongside Chinese (Chan and Siu 2016), Portuguese (Antunes, Silva, and Oliveira 2018), Greek (Polemikou, Zartaloudi, and Nikitas Polemikos 2019), Latvian (Grasmane, Raščevskis and Pipere 2022) or New Zealand (O’Sullivan and Lindsay 2022). Moreover, the studies in question supported the notion that personality/spirituality can use SI to achieve certain goals (Skrzypińska et al. 2019). Findings demonstrated that SI is helpful in gaining resilience or coping with the transgression through forgiveness, but not necessarily in enhancing wellbeing (Mróz, Kaleta, and Skrzypińska 2021; O’Sullivan and Lindsay 2022).
Grasmane, Ina, Vitālijs Raščevskis, and Anita Pipere. 2022. ‘Primary Validation of Children Spiritual Intelligence Scale in a Sample of Latvian Elementary School Pupils. International Journal of Children's Spirituality 27 (2), 97-112. https://doi.org/10.1080/1364436X.2022.2043833.
O’Sullivan, Laura, and Nicole Lindsay. 2022. ‘The Relationship between Spiritual Intelligence, Resilience, and Well-being in an Aotearoa New Zealand Sample’. Journal of Spirituality in Mental Health, 1-21. https://doi.org/10.1080/19349637.2022.2086840.
Polemikou, Anna, Eirini Zartaloudi, and Nikitas Polemikos. 2019. ‘Development of the Greek Version of the Spiritual Intelligence Self-Report Inventory-24 (KAPN): Factor Structure and Validation’. Mental Health, Religion & Culture 22 (10): 1033-1047. https://doi.org/10.1080/13674676.2019.1692195.
- On p. 4, 2.2 Participants and procedure, it would be better if a brief definition on the Declaration of Helsinki could be offered.
The information about The Declaration of Helsinki has been completed.
- On p. 5, 2.3.1, the psychometric property of the AB5C in the present study is better to be provided.
We added information about Cronbach’s alpha for AB5C in the present stud.
- On p.5, 2.4 Data Analysis, statistical tests used in the study are suggested to be described in this section.
All statistical tests have been described.
- On p. 6-7, a table may be used to clearly and systematically present the mediation effect of the four subscales of SI on the association between personality and decisional and emotional forgiveness.
We have added two tables with mediation effects of SI (4 subscales) on the relationship between personality and both forgiveness - Table 2 - emotional forgiveness, and Table 3 - decisional forgiveness. We have also included non-significant results.
- On p. 8, “The latter might require more effort……a balance of losses and profits “at 0”.” It would be helpful if more explanation could be provided for these two statements.
We have completed the explanation of the first sentence (below) and we removed the second because it seems confusing and is equal to the next phrase.
Approach motivation towards the offender is more complicated than just reducing negative motives which usually comes with time (McCullough, Fincham, and Tsang 2003).
- On p. 8, “In the future, when designing new research……can facilitate adaptations in life”. More empirical evidence could be provided to support this argument.
We gave the examples of such studies:
Sánchez-Álvarez, Nicolas, Natalio Extremera, and Pablo Fernández-Berrocal, 2016. ‘The Relation between Emotional Intelligence and Subjective Well-being: A Meta-analytic Investigation’. The Journal of Positive Psychology 11 (3): 276-285. https://doi.org/10.1080/17439760.2015.1058968.
Trigueros, Rubén, Ana M. Padilla, José M. Aguilar-Parra, Patricia Rocamora, María J. Morales-Gázquez, and Remedios López-Liria. 2020. ‘The Influence of Emotional Intelligence on Resilience, Test Anxiety, Academic Stress and the Mediterranean Diet. A Study with University Students’. International Journal of Environmental Research and Public Health 17 (6): 2071. http://dx.doi.org/10.3390/ijerph17062071.
Zeidner, Moshe, Gerald Matthews, and Richard D. Roberts. 2006. ‘Emotional intelligence, coping, and adaptation’. In J. Ciarrochi, J. Forgas, & J.D. Mayer (Eds.), Emotional intelligence in everyday life: A scientific inquiry (2nd edn., pp. 100–125). Philadelphia, PA: Psychology Press.
- On p. 9, the possible limitation of having many more female subjects then male subjects could be discussed.
We have completed the limitation section with the information about gender imbalance as follows:
Third, our sample was mostly female. Despite the fact that previous research has not demonstrated gender differences in spiritual intelligence (Pant and Srivastava, 2019), or has been inconsistent with forgiveness (Cabras et al. 2022; Kaleta and Mróz 2022), it is worthwhile for future research to address the gender imbalance, and to examine whether there are differences in the relationship between spiritual intelligence and forgiveness in females and males.